# Deep transfer learning-based bird species classification using mel spectrogram images

**Mrinal Kanti Baowaly**[1☯], **Bisnu Chandra Sarkar**[1☯], **Md. Abul Ala Walid**[2,3], **Md. Martuza Ahamad**[1], **Bikash Chandra Singh**[4], **Eduardo Silva Alvarado**[5,6,7], **Imran Ashraf**[8]*, **Md. Abdus Samad**[8]*

**1** Department of Computer Science and Engineering, Bangabandhu Sheikh Mujibur Rahman Science and Technology University, Gopalganj, Bangladesh, **2** Department of Computer Science and Engineering, Khulna University of Engineering and Technology, Khulna, Bangladesh, **3** Department of Data Science, Bangabandhu Sheikh Mujibur Rahman Digital University, Gazipur, Bangladesh, **4** School of Cybersecurity, Old Dominion University, Norfolk, VA, United States of America, **5** Universidad Europea del Atlántico, Santander, Spain, **6** Universidad Internacional Iberoamericana, Campeche, México, **7** Universidad de La Romana, La Romana, República Dominicana, **8** Department of Information and Communication Engineering, Yeungnam University, Gyeongsan-si, Gyeongsangbuk-do, South Korea

☯ These authors contributed equally to this work.
* imranashraf@ynu.ac.kr (IA); masamad@yu.ac.kr (MAS)

**Data Availability Statement:** The datasets utilized in this article were obtained from "BirdCLEF 2023: Identify bird calls in soundscapes" webpage, which

## Abstract

The classification of bird species is of significant importance in the field of ornithology, as it plays an important role in assessing and monitoring environmental dynamics, including habitat modifications, migratory behaviors, levels of pollution, and disease occurrences. Traditional methods of bird classification, such as visual identification, were time-intensive and required a high level of expertise. However, audio-based bird species classification is a promising approach that can be used to automate bird species identification. This study aims to establish an audio-based bird species classification system for 264 Eastern African bird species employing modified deep transfer learning. In particular, the pre-trained EfficientNet technique was utilized for the investigation. The study adapts the fine-tune model to learn the pertinent patterns from mel spectrogram images specific to this bird species classification task. The fine-tuned EfficientNet model combined with a type of Recurrent Neural Networks (RNNs) namely Gated Recurrent Unit (GRU) and Long short-term memory (LSTM). RNNs are employed to capture the temporal dependencies in audio signals, thereby enhancing bird species classification accuracy. The dataset utilized in this work contains nearly 17,000 bird sound recordings across a diverse range of species. The experiment was conducted with several combinations of EfficientNet and RNNs, and EfficientNet-B7 with GRU surpasses other experimental models with an accuracy of 84.03% and a macro-average precision score of 0.8342.

## Introduction

Bird species classification using deep learning and audio data is a rapidly evolving field with numerous applications in bioacoustics, ecology, ornithology, and conservation [1]. Bird

is freely accessible for all scientists and investigators to conduct experiments and can be accessed through the website: https://www.kaggle.com/competitions/birdclef-2023/data.

**Funding:** This study was funded by the European University of Atlantic. the funders had no role in study design, data collection and analysis, decision to publish, or preparation of the manuscript.

**Competing interests:** The authors have declared that no competing interests exist.

species are excellent indicators of environmental quality [2]. Researchers and conservationists can better understand population trends, migratory patterns, and ecosystem health with precise classifications of bird species. The relationship between bird species and their population trends, migratory patterns, and ecosystem health is essential to develop efficient conservation plans and prevent the extinction of threatened bird species [3].

The world is facing a biodiversity loss crisis due to human activities [4], such as environmental pollution, climate change, and habitat destruction. As a result, there is a pressing need to monitor and conserve species as indicators of biodiversity. Birds are critical indicators of environmental changes [5], such as changes in habitat, migration patterns, pollution, and disease outbreaks. Therefore, the conservation of bird species is critical to understanding environmental changes and preserving biodiversity worldwide. In particular, Eastern Africa is home to numerous bird species that play a vital role in the ecosystem and contribute to the cultural and economic value of the region [6]. However, monitoring and conserving these species is challenging due to the vast and diverse region, rugged terrain, and complex vocalizations. Traditional monitoring methods are often costly, time-consuming, and limited in data availability, particularly for rare and endangered species. However, these methods can be optimized using acoustic sensors [7]. In combination with deep learning techniques, wireless acoustic sensor networks present a novel approach to the classification of bird species [8, 9]. The wireless acoustic sensor networks in combination with deep learning approach uses audio recordings of bird vocalizations to identify species based on their unique acoustic patterns. It overcomes the limitations of visual identification [10] and enables the analysis of bird vocalizations in dense foliage. The wireless acoustic sensor networks in combination with deep learning method provide a noninvasive, cost-effective, and efficient means of continuous monitoring, even in vast and complex regions like Eastern Africa. There have been numerous studies conducted on the classification of audio based on raw audio signals [11, 12]. The raw audio signal can be transformed into images of mel-spectrograms that potentially have a significant impact on the analysis. Mel spectrogram images offer a valuable approach for classifying bird species through audio data due to their ability to represent frequency patterns effectively. By breaking down audio signals into distinct frequency components over time, mel-spectrograms serve as a powerful feature extraction technique [13]. Mel-spectrograms also provide a visually interpretable representation, aiding in both model validation and human analysis of acoustic characteristics. Furthermore, their dimensionality reduction compared to raw waveforms facilitates computational efficiency. This approach has demonstrated success in various audio classification tasks, making it a reliable choice for bird species identification. Authors in [14] shows that mel spectrogram images perform better as pre-processing for audio classification than raw audio data. The use of mel spectrogram images further enhances the accuracy and generalizability of classification models.

For image classification problems, some pre-trained Convolutional neural network (CNN) models such as VGG16 [15], Resnet [16], and EfficientNet [17] show outstanding performance. Pre-trained models are thus applicable to perform audio classification tasks like music genre categorization [18], environmental sound classification [19], and owl classification [20] which is based on mel spectrogram images derived from audio signals. EfficientNet is the most lightweight and has state-of-the-art performance among the aforementioned pre-trained models. As converting raw audio data to mel spectrogram images provides a better pre-processing step for audio data, there are several explorations towards classifying bird species using mel spectrogram images with CNN architectures [21, 22].

CNN models that exhibit greater performance generally possess broader architectures with encompass a vast number of parameters, necessitating a substantial volume of data for effective training and optimal performance [23]. In this particular context, to lessen the duration

required for training and to mitigate the challenges associated with training using extensive data sets, a number of methodologies employ the technique of transfer learning with updating weights of pre-trained models by fine-tuning [24–27]. However, in order to effectively adopt the temporal relationships included in sequence data, RNNs such as LSTM [28] and GRU [29] have demonstrated notably superior performance. Hybrid models leverage the inherent advantages of convolutional and recurrent neural networks to effectively acquire knowledge from temporal or sequential data. Convolutional layers are employed to extract localized patterns at each temporal instance, followed by integrating the acquired representations over numerous temporal instances by utilizing a recurrent component. Recent studies described in the literature [30–32] have demonstrated that hybrid models exhibit enhanced performance compared to baseline single CNN models across a range of sound detection tasks.

This study aims to develop robust models for accurately identifying and classifying Eastern African bird species based on their vocalizations, represented as mel spectrogram images. The advantages of transfer learning from a larger image classification model embedded with RNNs is applied to achieve to develop robust models for accurately identifying and classifying Eastern African bird species. To the best of our knowledge, there is a lack of research conducted utilizing this dataset, which encompasses a substantial number of species groups. By leveraging transfer learning, specifically through pre-trained models like EfficientNet, the accuracy and efficiency of our classification model can be improved. Moreover, including RNN layers allows us to effectively capture the sequential dependencies included in the mel spectrogram images. This integration improves the classification performance of our model, specifically for Eastern African bird species. In general, the main contributions of this study are listed as follows.

- The application of the mel spectrogram transformation is performed on a predetermined section of audio in order to produce mel spectrogram images.

- A novel fine tuned EfficientNet-B7-based architecture for classifying Eastern African bird species is proposed in this study.

- Experimenting several combinations of fine-tuned EfficientNet and RNN variants, including LSTM and GRU models.

- Conducting a comparative investigation and selecting a robust model for the accurate and efficient identification of Eastern African bird species.

The remainder of this paper is organized as follows. Section provides an overview of related work concerning bird species classification based on audio and deep learning techniques. Section details the proposed methodology, including a description of the BirdCLEF 2023 dataset, pre-processing steps, training hyperparameters, the environmental setting for training, proposed models, and the evaluation metrics used in our study. Section represents the experimental results and evaluation metrics. Section discusses the performance of different models employed in our study. Finally, Section concludes the article and discusses directions for future work.

## Related work

Implementing deep learning techniques, specifically CNN, has sparked notable interest in recognizing bird species through vocalizations. These recent developments are crucial to improving our knowledge of avian biodiversity and fostering the development of more effective conservation strategies. Authors in [8] pioneered the application of CNNs to identify bird species from audio recordings. The authors of the paper used a CNN with five convolutional layers and one dense layer. They processed the audio data by dividing it into a signal part

containing bird vocalizations and a noise part representing background noise. Spectrograms were computed for both parts and split into fixed-length segments. They applied their approach to the BirdCLEF 2016 bird identification task and achieved a mean average precision score of 0.686.

Furthermore, authors in [33] also approached the bird identification task with their custom CNN model. The authors used mel-spectrograms of processed audio data, applied data augmentation and normalization techniques, and segmented the recordings into fixed-length segments. In the BirdCLEF 2017 dataset, they achieved a mean average precision score of 0.605.

Hybrid models leverage convolutional and recurrent neural network (RNN) strengths to learn from temporal or sequence data. Some works are on CNN + RNN hybrid models [34, 35]. Authors in [35] experimented with state-of-the-art image classification methods based on transfer learning (ResNet18, ResNet50, VGG16) and obtained the highest accuracy of 61.9% with VGG16. They enhanced their model performance using the embedded hybrid CNN and RNN model, getting the highest accuracy of 67%. They used the Cornell Bird Challenge 2020 dataset. This dataset contains 264 classes of bird species. However, they used only 100 classes of bird species to handle class imbalance problems.

In [36], the authors also used the Cornell Bird Challenge 2020 dataset. They used all bird classes. The proposed model architecture for the CNN framework used residual learning and attention mechanisms to generate attention-aware features, which enhanced the overall accuracy of bird call identification. They got the highest accuracy, up to 64.37%.

Although substantial progress has been reported in classifying birds, there are still unexplored ways to maximize output. One of these avenues is the transfer of knowledge from larger pre-trained image classification models embedded with RNNs to process temporal data. Our novel approach investigates this avenue, contributing to the expanding body of knowledge and paving the way for enhanced bird monitoring, biodiversity assessment, and conservation efforts. In particular, our study uses the BirdCLEF 2023 dataset, and it is intended to resolve the limitations and expand on the work of previous researchers. This study represents the pioneering effort to investigate the BirdCLEF 2023 dataset to classify Eastern African birds.

## Materials and methods

This study aims to develop an accurate and efficient audio-based bird species classification system by using mel spectrogram images and deep learning techniques. The study is divided into several steps. First, the BirdCLEF2023 dataset is collected, which focuses on identifying bird calls in soundscapes, from Kaggle [37]. Subsequently, pre-processing steps are performed, which involve converting audio data into mel spectrogram images. In addition our dataset is split into training, testing, and validation sets and applied random upsampling on the training set to minority classes in order to solve class imbalance issues, followed by image augmentation techniques. In the next step, the fine-tuned EfficientNet network using the pre-processed dataset is trained. Additionally, the LSTM and GRU models are assembled with EfficientNet to explore the effectiveness of recurrent neural networks in the classification task. Finally, the model was optimized with experimenting several structure of top layers, identify the best-performing model, and thoroughly evaluate all the models developed. The entire process of the work is depicted in Fig 1.

### Data collection and pre-processing

In order to attain precise and effective categorization, the present study uses the BirdCLEF 2023 dataset (https://www.kaggle.com/competitions/birdclef-2023/data), which was specifically curated to classify avian species. Table 1 presents some basic dataset statistics. The

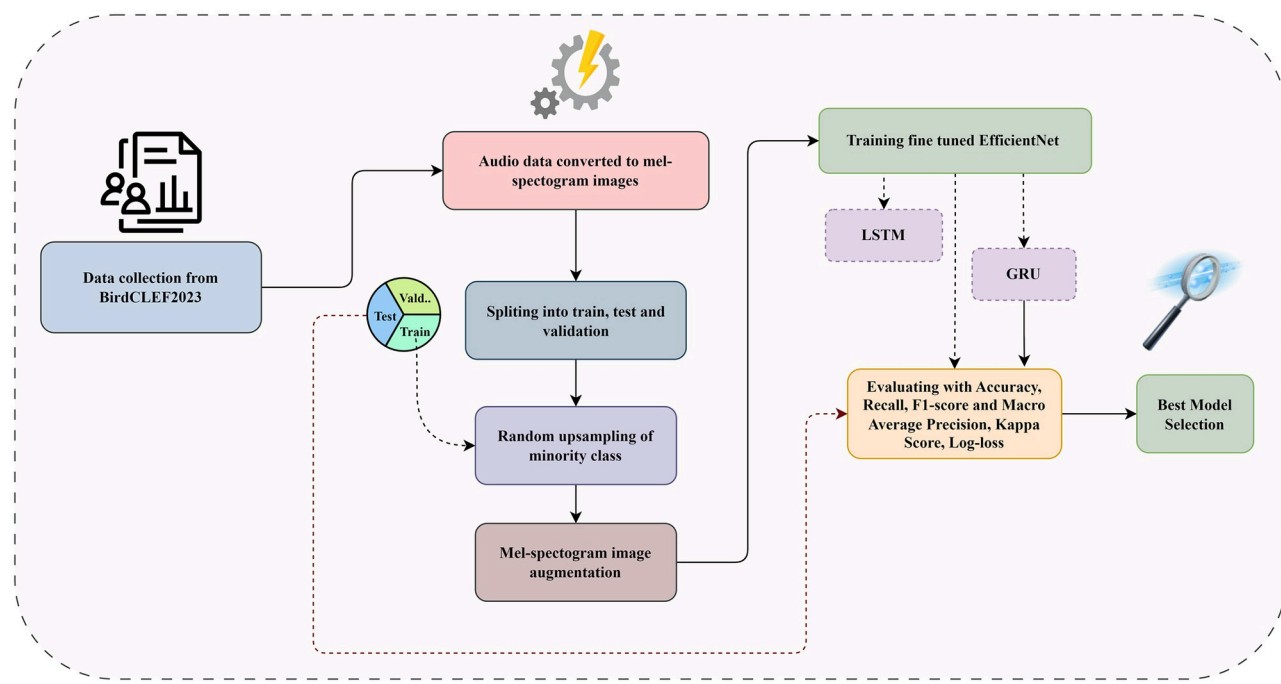

**Fig 1. The workflow of proposed method.**

collection consists of 16,753 audio signals, each with a sample rate of 32,000 Hz, representing audios from 264 distinct species.

However, the dataset presents class imbalance issues that require appropriate techniques to address them. Random minority upsampling was implemented to ensure sufficient representation of rare and endangered species in the training data. By overcoming class imbalance challenges and using deep learning techniques, the approach proposed in this study offers a versatile approach to monitoring and conserving bird populations, enabling a better understanding of environmental changes and supporting regional conservation efforts.

The dataset focuses on bird audio recordings and associated metadata. The main component of the data set is the "train_audio" directory, which contains a collection of short recordings of individual bird calls. These recordings were generously uploaded by users of xenocanto.org, a platform dedicated to sharing bird sounds. The audio files were down-sampled to 32 kHz, where applicable, to match the test set audio, and converted to the ogg format. In addition to the audio files, the dataset includes the "train_metadata.csv" file, which provides extensive metadata for the training data. One of the key fields in this file is the "primary_label," which represents the code for the bird species associated with each recording. Fig 2 shows the distribution of bird species' images.

To pre-process the dataset and facilitate subsequent analysis, the following steps were undertaken:

**Table 1. Basic statistics of dataset.**

| Items | Value |
|---|---|
| Number of bird sounds / audio recordings | 16753 |
| Number of bird species | 264 |
| Number of sampling rate of audios | 32000 Hz |

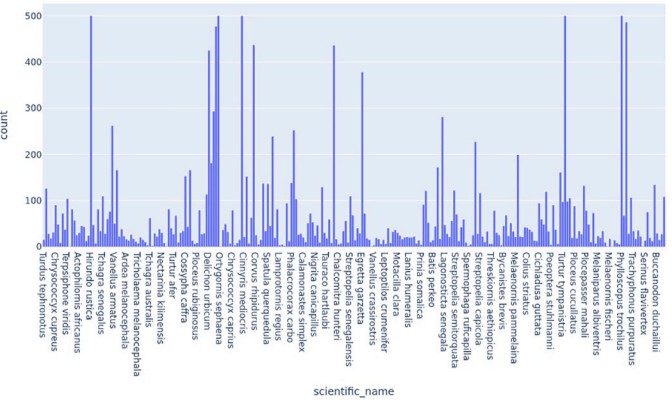

**Fig 2. Distribution of birds' species data.**

1. The files and their associated labels were collected from the database.

2. The audio data was partitioned into 5-second segments, and segments with durations shorter than 5 seconds were cyclically padded to meet the necessary duration.

3. The mel spectrogramtransformation was applied to each 5-second segment of audio data with the parameters given in Table 2.

4. The spectrograms were transformed into a decibel scale and then normalized to the uint8 image range. Fig 3 provides an example of the resulting spectrograms, showing the representation of an audio file following the proposed transformation process. Fig 3(a) to 3(f) depict mel spectrogram images of six bird species out of a total of 264 species.

5. The primary label was assigned to each of these created spectrogram images. The primary label denotes the predominant avian species or the auditory classification within the given time segment. The relation between the primary label and the corresponding spectrum images is crucial for training and classification purposes.

Following these pre-processing steps, the dataset was transformed into mel spectrogram images, with primary labels assigned to each interval, padding applied for minimal durations, and independent image creation. Pre-processing laid the foundation for further exploration, feature extraction, and model development for identifying and classifying bird species.

## Training hyper-parameters

The training hyper-parameters is a multiclass and supervised learning classification task. Therefore, the following loss function, binary cross-entropy, was calculated in each batch for

**Table 2. Table of parameters and values.**

| Parameter | Value |
| --- | --- |
| Samples Between Consecutive Windows | 2048 |
| Number of Mel Frequency Bands | 128 |
| Minimum frequency of the Mel Filter Bank (Hz) | 500 |
| Maximum frequency of the Mel Filter Bank (Hz) | 12500 |
| Centered | False |

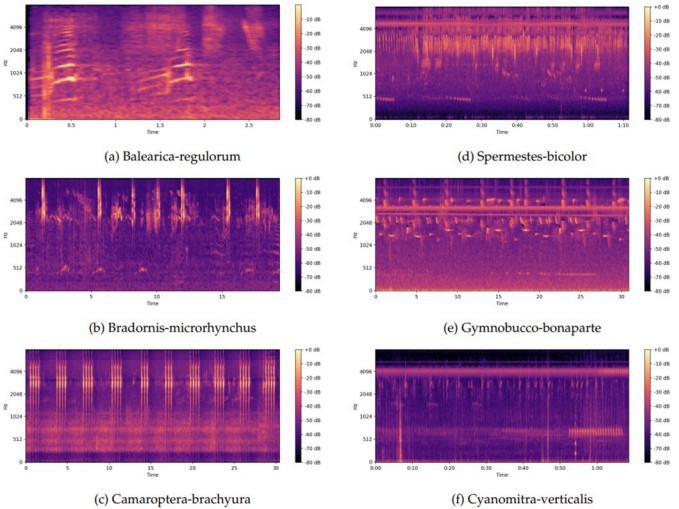

**Fig 3. Mel-spectogram images of some bird species audio.**

each class separately:

$$L(y_i, \hat{y}_i) = -[y_i \log(\hat{y}_i) + (1 - y_i) \log(1 - \hat{y}_i)] \tag{1}$$

where, $y_i$ is the ground truth label for the $i^{th}$ class, and $\hat{y}_i$ is the predicted probability generated by the model output for the $i^{th}$ class. In each training epoch, the mean loss was calculated by summing up the binary cross-entropy loss for each class and each batch and dividing it by the number of classes and batches. The mean loss was optimized during the epochs.

This study employed the 'Adam' optimizer for the training process, using an initial learning rate of $1 \times 10^{-4}$. Adam Optimizer was chosen for its ability to adapt to the adjustment of learning rates, contributing to more efficient and stable training. The Adam optimizer optimizes the model parameters during training by utilizing adaptive learning rates.

Following established practices in deep learning, For our experiments, a batch size of 128 is carefully selected. It is crucial to consider that smaller batch sizes might result in a slower convergence rate, but are often necessary when dealing with larger models due to memory limitations. On the contrary, larger batch sizes can lead to unstable graphics memory usage, potentially causing training algorithm failures. Therefore, our choice of batch size strikes a well-balanced approach between computational efficiency and model stability, ensuring a smooth and effective training process.

## Environmental setting

To ensure a high-quality implementation of our methodology, TensorFlow version 2.11.0 along with the TensorFlow I/O library version 0.29.0 and Keras-CV version 0.4.2 were effectively used. Using a GPU P100, model training was conducted in a Kaggle environment. The training procedure was supported by the computational environment provided by the Kaggle environment. GPU P100, in conjunction with Kaggle configuration, ensured optimal performance and efficient execution of computationally intensive training procedure tasks.

## Proposed models

The audio files were initially converted to mel spectrogram images, visually representing the frequency content of bird vocalizations over time. The conversion process allowed the extraction of relevant features from the audio data. To mitigate the problem of class imbalance, random minority upsampling procedures were used. The proposed technique helps to overcome the imbalance in the number of samples available for different species of birds, ensuring that each class was a sufficient representation in the training data. The three different model architectures are experimented in this study. Several considerations influenced our choice to employ these particular models. The choices are discussed in the following subsections:

**EfficientNet-B7.** The introduction of EfficientNet in the study [17] that revolutionized the field of model scaling in CNNs. Unlike traditional approaches that arbitrarily scaled network dimensions, EfficientNet introduced a novel method that uniformly scales all three dimensions (width, depth, and resolution) using a composite coefficient. Among the various members of the EfficientNet family, EfficientNet-B7 [17] stands out as the most significant variant, demonstrating exceptional performance on the ImageNet dataset and achieving new accuracy records. The EfficientNet-B7 model excels in computational efficiency and high accuracy.

It was maintained trainable for the final 200 levels and continues to freeze the first 612 layers for transfer learning from EfficientNet-B7. The intuition behind keeping trainable for the last 200 layers and keep freezing the first 612 layers during fine-tuning is the larger image classification model's learning pattern for the image classification task. The beginning layers of such models focus on lower-level patterns such as edges, textures, and basic shapes. On the other hand, the last layers learn complex object shapes, semantic concepts, and contextual information that spans a broader area. As in spectrogram image classifications, it also requires pattern recognition such as edges, textures, and basic shapes at the beginning, and the first several layers remain frozen. However, as in our problem, complex patterns are unseen by the larger image classification model; the last layers are kept trainable during fine-tuning to learn patterns of complex patterns of spectrogram images for bird species classification. Fine-tuning allows us to achieve a considerable improvement in performance over our baseline EfficientNet-B7 model.

**EfficientNet-B7 embedded with LSTM.** Using LSTM [28] in modeling sequential data has become common practice in neural networks. LSTMs, a variant of RNNs, excel at capturing long-range dependencies in data, making them particularly suitable for tasks such as natural language processing, speech recognition, and machine translation. These networks were initially introduced by Sepp Hochreiter and Jürgen Schmidhuber in 1997 as a solution to the vanishing-gradient problem that can arise when training RNNs on lengthy data sequences. LSTMs employ a gated mechanism to regulate the flow of information throughout the network. This mechanism comprises three gates: the input gate, the forget gate, and the output gate. The input gate determines the amount of current input that should be stored in the cell state. In contrast, the forget gate determines the degree to which the previous cell state should be disregarded. Lastly, the output gate determines to what extent the cell state should be passed to the subsequent layer of the network. By employing these gates, LSTMs can effectively control the information flow, mitigating the vanishing gradient problem, and enabling the modeling of long-term dependencies.

The rationale behind incorporating an LSTM with EfficientNet-B7 is to leverage the ability to capture temporal patterns. The use of LSTM with EfficientNet-B7 network is particularly relevant in scenarios where specific audio characteristics, such as temporal patterns exhibited by birds, cannot be easily learned by the pre-trained CNN architecture of EfficientNet alone. By integrating an LSTM, the model can effectively recognize and understand these temporal patterns, enhancing its overall performance in bird audio recognition tasks. A better performance was achieved than the baseline EfficientNet-B7 architecture by applying the following architecture.

**EfficientNet-B7 embedded with GRU.** GRUs [29] were introduced in 2014 by Kyunghyun Cho *et al.* as a variant of RNNs. GRUs share similarities with LSTMs but have fewer parameters, resulting in faster training times. Additionally, GRUs exhibit a simpler architecture, which enhances their comprehensibility and ease of implementation. GRUs employ a gated mechanism to regulate the flow of information within the network.

The gated mechanism with GRUs comprises two gates: the update gate and the forget gate. The update gate determines the proportion of the previous hidden state that should be retained. In contrast, the forget gate determines the amount of the previous hidden state that should be ignored. The new hidden state is computed as a linear combination of the previous and current input, with the update gate controlling the weights.

GRUs have gained popularity as an option for RNNs because of their advantageous characteristics, such as speed, efficiency, and ease of understanding. To use the advantages of GRU's over LSTM, our fine-tuned EfficientNet-B7 architecture is embedded with GRU. By applying the following architecture, the better performance was noticed by comparing to baseline EfficientNet-B7 architecture and slightly better than EfficientNet-B7 embedded with LSTM. The implementation of all the codes of this study is available in the website link: https://zenodo.org/records/10732565.

## Evaluation metrics

To evaluate the proposed models used in this study, a set of popular evaluation metrics have been utilized for the classification task, including accuracy, macro average precision, recall, and F1 score. Most metrics depend on the values of the confusion matrix, that is, true positive (TP), true negative (TN), false positive (FP), and false negative (FN). A brief description of those metrics is as follows.

- **Accuracy**: One way to judge the performance of a classification model is by its accuracy. Accuracy can be defined informally as the percentage of correct predictions made by our model [38]. The Eq (2) to calculate the accuracy score of a binary classification model is as follows.

$$Accuracy = \frac{TP + TN}{TP + TN + FP + FN} \tag{2}$$

- **Precision and Macro Averaged Precision**: The precision score indicates the percentage of correct affirmative identifications that were [39]. Another way to describe the precision score is macro-averaged precision (mAP); it is calculated after taking class-wise precision scores and then averaging those scores. Eqs (3) and (4) equation are used to calculate precision and macro averaged precision.

$$Precision(P) = \frac{TP}{TP + FP} \tag{3}$$

$$mAP = \frac{P_1 + P_2 + \ldots + P_n}{n} \tag{4}$$

where, there are n classes, 1, 2, . . . n, and the precision scores are $P_1, P_2, \ldots P_n$, respectively.

- **Recall/Sensitivity**: In what percentage were true positives properly diagnosed, is based on the recall to define it [40]. The Eq (5) is used to perform the following computation:

$$Recall/Sensitivity = \frac{TP}{TP + FN} \tag{5}$$

- **F1 Score**: In contrast to accuracy, which focuses on the overall performance of the model, the F1 score evaluates the predictive capacity of the model by delving into its performance in each class. Model precision and recall are two separate criteria. However, the F1 score averages them together [38].

$$F1 = \frac{2 \times Precision \times Recall}{Precision + Recall} = \frac{2 \times TP}{2 \times TP + FP + FN} \tag{6}$$

- **Cohen's kappa score**: The Kappa coefficient is a statistical measure that compares the agreement between observed accuracy and expected accuracy. The Kappa coefficient is a metric used to evaluate the effectiveness of a classification model by comparing its performance to that of a random classifier. The metric Kappa can be computed using (7), where TACC represents total accuracy and RACC represents random accuracy.

$$K = \frac{TACC - RACC}{1 - RACC} \tag{7}$$

- **Log loss**: The log-loss metric quantifies the uncertainty of a probabilistic approach by evaluating its accuracy in predicting true labels. A log-loss score that is relatively low indicates a high level of accuracy in the forecast. The application of (8) enables the calculation of the log loss of a model [39] where x represents the level of the destination variable and p(x) denotes the projected probability associated with the specific value of interest.

$$L - loss = -\frac{1}{N} \sum_{i=1}^{N} y_i \cdot \log(p(x_i)) + (1 - x_i) \cdot \log(1 - p(x_i)) \tag{8}$$

## Results

Our results were evaluated locally by performing a dataset split into 80% for training, 10% for testing, and 10% for validation. To preserve the original label distribution, the scikit-learn's 'train_test_split' function was utilized, which handles the class distribution by stratifying the splits. In this work, the three deep neural network models were applied: EfficientNet-B7, EfficientNet-B7 with LSTM, and EfficientNet-B7 with GRU. The performances of our models were evaluated using four evaluation metrics: accuracy, recall, F1 score, and macro-average precision. The models were trained for multiple epochs, and their specific results are discussed in the following subsections:

### Training results of EfficientNet-B7

The baseline model EfficientNet-B7 was trained for 70 epochs, resulting in a training accuracy of 84.10% and a validation accuracy of 81.96%. On the test dataset, the model achieved an

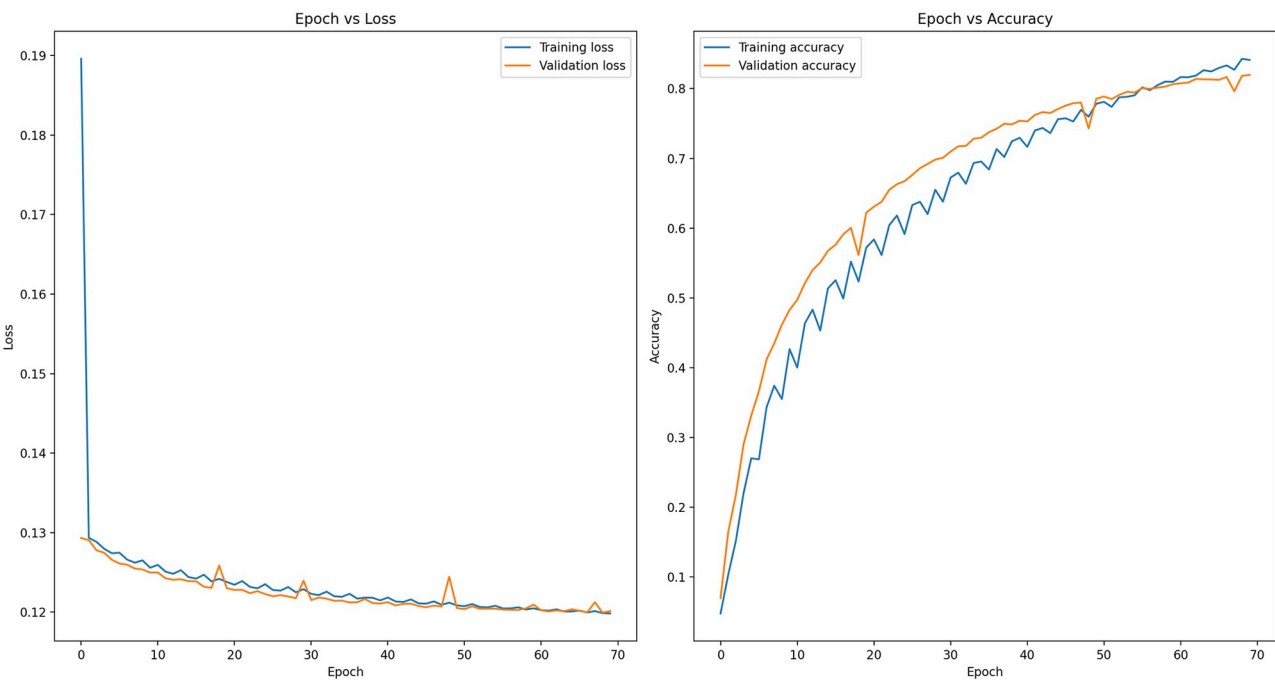

**Fig 4. Accuracy and loss of fine-tuned EfficientNet-B7 transfer learning model.**

accuracy of 81.82%. Furthermore, the model performance metrics on the test dataset were as follows: recall of 0.7129, F1 score of 0.7424, macro-average precision of 0.8048, Cohen's kappa score of 0.8152, and log loss of 3.34.

In Fig 4, epoch-wise train/valid loss and accuracy scores in epoch are depicted. The left side of the figure shows the train/valid loss decreasing over the epochs. On the contrary, the right side shows the train/valid accuracy increasing over the epochs. Once the model reached approximately 70 epochs, its loss neared convergence, indicating that further training would lead to overfitting. Therefore, the training at that point was concluded.

## Training results of EfficientNet-B7 + LSTM

For the later model, EfficientNet-B7 with LSTM was implemented. The model was trained for 69 epochs, resulting in a training accuracy of 89.56% and a validation accuracy of 84.03%. On the test dataset, the model achieved an accuracy of 83.67%. Additionally, the model's performance metrics on the test dataset were as follows: recall of 0.7552, F1 score of 0.7816, macro-average precision of 0.8326, Cohen's kappa score of 0.8339, and log loss of 3.67.

In Fig 5, epoch-wise train/valid loss and accuracy scores are depicted in the epoch. The left side of the figure shows the train/valid loss decreasing over the epochs. On the contrary, the right side shows the train/valid accuracy increasing over the epochs. After completing 69 epochs, the model's loss demonstrates a tendency towards convergence. Subsequently, a minor overfitting of the model occurs. Our work was engaged in training until the occurrence of overfitting was noticed.

The model consistently outperforms the base model in all aspects. Specifically, it shows a better performance of 3% accuracy and macro-average precision than the base model. Furthermore, the recall and F1 scores demonstrate a remarkable improvement of almost 4% over the

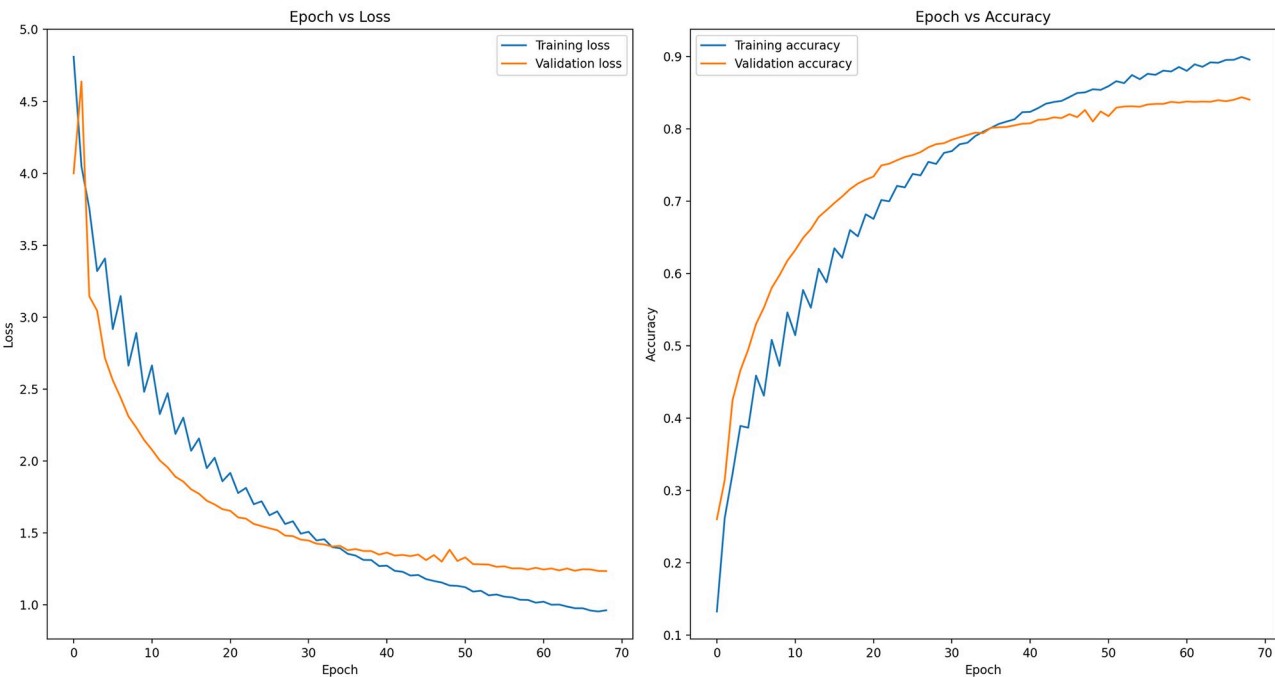

**Fig 5. Accuracy and loss of fine-tuned EfficientNet-B7 embedded with LSTM.**

base model. These findings highlight the enhanced capabilities and effectiveness of this model compared to the baseline.

### Training results of EfficientNet-B7 + GRU

In our third model, EfficientNet-B7 with GRU was implemented. The model was trained for 70 epochs, resulting in a training accuracy of 90.80% and a validation accuracy of 84.61%. On the test dataset, the model achieved an accuracy of 84.03%. Additionally, the model's performance metrics on the test dataset were as follows: recall of 0.7738, F1 score of 0.7924, macro average precision of 0.8342, Cohen's kappa score of 0.8376, and log loss of 3.65. In Fig 6, epoch-wise train/valid loss and accuracy scores are depicted. The left side of the figure shows the train/valid loss decreasing over the epochs. On the contrary, the right side shows the train/valid accuracy increasing over the epochs. After completing 70 epochs, the model loss approached convergence, but a slight indication of overfitting became apparent. Therefore, it was decided to conclude the training at that stage.

The performance of the model is slightly better than the EfficientNet-B7 + LSTM model's accuracy, macro-average precision, and F1 score. However, it exhibits significant improvements in recall. These results highlight its superior capabilities to capture relevant information and achieve higher precision on average.

### Discussions

We mentioned earlier that we experimented with three models. We now wish to proceed with the analysis of the findings of our investigation, compare the results, and find the best result. The comparative results of the three proposed models on the test data are shown in Table 3. These models were evaluated using accuracy, recall, F1 score, and macro average precision.

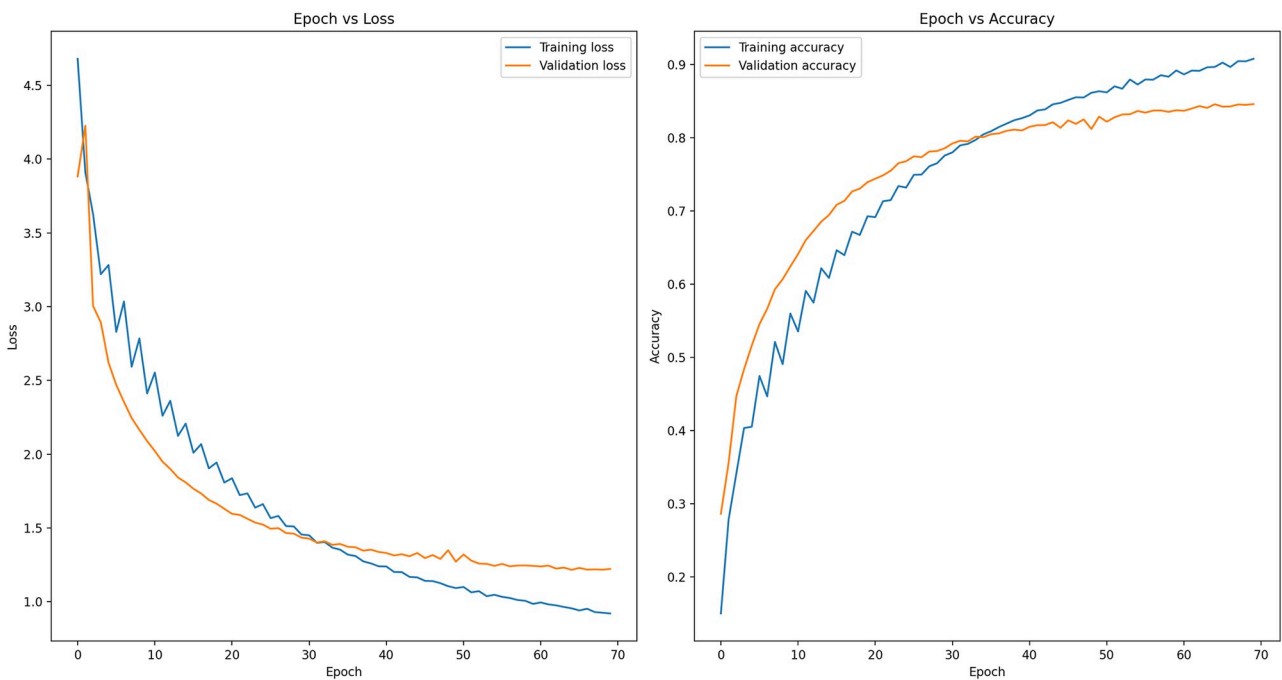

**Fig 6. Accuracy and loss of fine-tuned EfficientNet-B7 embedded with GRU.**

The initial model, EfficientNet-B7, demonstrates a moderate level of accuracy at 0.8182, indicating that it correctly identifies bird species in approximately 81.82% of cases. However, it falls short in terms of recall (0.7129), F1 score (0.7424), macro-average precision (0.8048), Cohen's kappa score (0.8152), and log loss (3.34). These metrics suggest that while Efficient-Net-B7 can accurately classify some species of birds, it can struggle to identify specific species correctly.

However, adding LSTM to the EfficientNet-B7 model significantly improves its overall performance. The EfficientNet-B7 + LSTM model shows notable enhancements in accuracy (83.67%), recall (0.7552), F1 score (0.7816), macro-average precision (0.8326) and Cohen's kappa score (0.8339). These improvements indicate that the incorporation of LSTM helps capture temporal dependencies in audio data, resulting in better classification results.

Similarly, incorporating GRU also improves the overall performance of the model. The EfficientNet-B7 + GRU model achieves an accuracy of 84.03%, slightly exceeding the LSTM model. It also shows comparable improvements in EfficientNet-B7 + GRU model in recall (0.7738), F1 score (0.7924), macro average precision (0.8342) and Cohen's kappa score (0.8376).

Therefore, it is evident that adding GRU to the EfficientNet-B7 architecture effectively models temporal dependencies, contributing to improved classification results.

**Table 3. Comparative results of different models on test data.**

| Technique | Accuracy | Macro average precision | Recall | F1-score | Cohen's kappa score | Log loss |
|---|---|---|---|---|---|---|
| EfficientNet-B7 | 81.82% | 0.8048 | 0.7129 | 0.7424 | 0.8152 | 3.34 |
| EfficientNet-B7 + LSTM | 83.67% | 0.8326 | 0.7552 | 0.7816 | 0.8339 | 3.67 |
| EfficientNet-B7 + GRU | 84.03% | 0.8342 | 0.7738 | 0.7924 | 0.8376 | 3.65 |

The EfficientNet-B7 + LSTM and EfficientNet-B7 + GRU models outperform the stand-alone EfficientNet-B7 model. They achieve higher precision, recall, F1 score, and macro-average precision, indicating their effectiveness in capturing relevant patterns specific to the Eastern African bird species classification task. Incorporating LSTM and GRU enables the models to leverage temporal dependencies in the audio data, resulting in more accurate and reliable classification.

## Conclusions

This study introduces a robust audio-based bird species classification system designed explicitly for Eastern African birds. The system achieves an impressive accuracy of 84.03% using the BirdCLEF 2023 dataset. It consists of two main components: a mel spectrogramimage generator and a hybrid deep learning classifier. The mel spectrogramimage generator converts audio recordings of bird vocalizations into mel spectrogram images, which are then used as input for the deep learning classifier. The classifier combines the strengths of CNNs and RNNs, where CNNs extract local patterns from the mel spectrogram images and RNNs capture temporal dependencies within the audio data. By incorporating augmented techniques and exploring various deep-learning architectures, this study significantly improves the accuracy and utility of bird species classification. As a result, it makes a valuable contribution to bird conservation, environmental monitoring, and wildlife research endeavors.

Future work will focus on incorporating more effective augmentation techniques to improve classification accuracy. Alternative deep learning architectures can also be employed to explore additional improvements. Additionally, the system will be evaluated on other bird species and various environmental conditions to ensure its versatility and generalizability.

## Author Contributions

**Conceptualization:** Mrinal Kanti Baowaly, Bisnu Chandra Sarkar.

**Data curation:** Bisnu Chandra Sarkar, Md. Abdus Samad.

**Formal analysis:** Md. Abul Ala Walid, Md. Martuza Ahamad.

**Funding acquisition:** Eduardo Silva Alvarado.

**Investigation:** Eduardo Silva Alvarado, Imran Ashraf.

**Methodology:** Mrinal Kanti Baowaly, Bisnu Chandra Sarkar, Md. Abul Ala Walid.

**Project administration:** Mrinal Kanti Baowaly, Eduardo Silva Alvarado.

**Resources:** Bikash Chandra Singh.

**Software:** Md. Martuza Ahamad, Bikash Chandra Singh.

**Supervision:** Md. Abdus Samad.

**Validation:** Imran Ashraf, Md. Abdus Samad.

**Visualization:** Md. Martuza Ahamad, Bikash Chandra Singh.

**Writing – original draft:** Mrinal Kanti Baowaly, Bisnu Chandra Sarkar.

**Writing – review & editing:** Md. Abul Ala Walid, Imran Ashraf.

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
