## [Decision Letter · Decision Letter 0]

12 Feb 2024

PONE-D-23-37709Deep Transfer Learning-Based Bird Species Classification Using Mel Spectrogram ImagesPLOS ONE

Dear Dr. Samad,

Thank you for submitting your manuscript to PLOS ONE. After careful consideration, we feel that it has merit but does not fully meet PLOS ONE’s publication criteria as it currently stands. Therefore, we invite you to submit a revised version of the manuscript that addresses the points raised during the review process. **Your manuscript has been evaluated by two reviewers, and their comments are appended below.** **Both reviewers have raised concerns about the overall clarity and communication of methodological details for this study. Please ensure you address each of the reviewers' comments, as well as the editorial requirements regarding availability of code, when revision your manuscript files.** **Please be aware that significant revisions will be required to address these concerns, and your manuscript may be rejected if your revisions do not adequately address these comments. Publications in PLOS ONE must report experiments and other analyses that are performed to a high technical standard and described in sufficient detail, and must be presented in an intelligible fashion.**

We look forward to receiving your revised manuscript.

Kind regards,

Hugh Cowley

Staff Editor

PLOS ONE

Journal Requirements:

This study was funded by the European University of Atlantic.

5. We notice that your supplementary figures are uploaded with the file type 'Figure'. Please amend the file type to 'Supporting Information'. Please ensure that each Supporting Information file has a legend listed in the manuscript after the references list.

Reviewers' comments:

Reviewer's Responses to Questions

**Comments to the Author**

1. Is the manuscript technically sound, and do the data support the conclusions?

Reviewer #1: No

Reviewer #2: Yes

2. Has the statistical analysis been performed appropriately and rigorously? 

Reviewer #1: No

Reviewer #2: Yes

3. Have the authors made all data underlying the findings in their manuscript fully available?

Reviewer #1: No

Reviewer #2: Yes

4. Is the manuscript presented in an intelligible fashion and written in standard English?

Reviewer #1: No

Reviewer #2: Yes

5. Review Comments to the Author

Reviewer #1: Authors worked on Bird Species and the manuscript have major challenges

1. In abstract, the authors mentioned background in present form. Is that current challenges of the technique of previous challenge?

2. "In this study, our aim is to develop an audio-based bird species" statement is not upto the mark.

3. Authors must not use words like "we", "me", "I", "if" etc.

4. " In particular, we explore using EfficientNet, a pre-trained convolutional neural network (CNN) originally trained using mel spectrogram images derived from bird sound recordings." structure is not upto the mark.

5. somewhere they mentioned 264 bird species and some where 17,000 bird sounds recording. Statement is not clear.

6. "surpasses competing models with an accuracy of 84.03\\% and a macro-average precision score of 0.8342." typo error in this statement.

7. Figures are missing only captions are there.

It seems the authors were in hasty while writing the manuscript. They are suggested to read the whole manuscript carefully before submission.

Reviewer #2: This study aims to develop an audio-based bird species classification system for 264 bird species in Eastern Africa using deep transfer learning, particularly employing EfficientNet combined with Gated Recurrent Unit (GRU) and Long Short-Term Memory (LSTM) networks.

I think that various major deficiencies in the study should be corrected.

- The parameters used to obtain the Mel-spectrogram should be given.

- Why Mel-spectrogram features are preferred should be explained in more detail.

- The results need to be compared across literatures.

- perform k-fold cross-validation

6. PLOS authors have the option to publish the peer review history of their article (what does this mean?). If published, this will include your full peer review and any attached files.

Reviewer #1: **Yes: **Dr. Rakesh Kumar

Reviewer #2: No

---

## [Author Response · Author response to Decision Letter 0]

15 Apr 2024

We have included revised cover letter during this submission. 

In addition we have attached a file that contains our responses to the reviewers.

---

## [Decision Letter · Decision Letter 1]

5 Jun 2024

Deep Transfer Learning-Based Bird Species Classification Using Mel Spectrogram Images

PONE-D-23-37709R1

Dear Dr. Samad,

We’re pleased to inform you that your manuscript has been judged scientifically suitable for publication and will be formally accepted for publication once it meets all outstanding technical requirements.

Kind regards,

Asadullah Shaikh, Ph.D.

Academic Editor

PLOS ONE

Additional Editor Comments (optional):

Reviewers' comments:

Reviewer's Responses to Questions

**Comments to the Author**

1. If the authors have adequately addressed your comments raised in a previous round of review and you feel that this manuscript is now acceptable for publication, you may indicate that here to bypass the “Comments to the Author” section, enter your conflict of interest statement in the “Confidential to Editor” section, and submit your "Accept" recommendation.

Reviewer #1: All comments have been addressed

Reviewer #2: All comments have been addressed

2. Is the manuscript technically sound, and do the data support the conclusions?

Reviewer #1: Yes

Reviewer #2: Yes

3. Has the statistical analysis been performed appropriately and rigorously? 

Reviewer #1: Yes

Reviewer #2: Yes

4. Have the authors made all data underlying the findings in their manuscript fully available?

Reviewer #1: Yes

Reviewer #2: Yes

5. Is the manuscript presented in an intelligible fashion and written in standard English?

Reviewer #1: Yes

Reviewer #2: Yes

6. Review Comments to the Author

Reviewer #1: Authors worked on "Deep Transfer Learning-Based Bird Species Classification Using mel spectrogram images"

authors have addressed all the questions raised. No more questions are left to ask.

Reviewer #2: The revision was done properly. All issues have been addressed. I believe it can be accepted as it is.

7. PLOS authors have the option to publish the peer review history of their article (what does this mean?). If published, this will include your full peer review and any attached files.

Reviewer #1: **Yes: **Dr. Rakesh Kumar

Reviewer #2: No
